# The total gut mucosal and fecal bacterial load increases in successful treatment of inflammatory bowel disease with infliximab

Rebecka Ventin-Holmberg,[1,2] Julia Eriksson,[2] Anja Eberl,[3] Taina Sipponen,[3] Eija Nissilä,[1,4,5] Päivi Saavalainen[2,5]

**ABSTRACT**  Inflammatory bowel diseases (IBD), including Crohn's disease (CD) and ulcerative colitis (UC), are chronic inflammatory gastrointestinal disorders linked to genetic predisposition and environmental factors. The gut microbiota, composed of various microorganisms, plays a crucial role in IBD, as reduced anaerobic bacteria and short-chain fatty acid (SCFA) producers are associated with predisposition to IBD. There is no cure for IBD, but the treatment aims for mucosal healing including conventional treatment and biological therapies such as infliximab (IFX). IFX, a tumor necrosis factor alpha (TNF-α) blocker, effectively reduces inflammation, but around 50% of patients do not achieve long-term remission. Fecal samples were collected from 70 patients with IBD (24 CD, 44 UC, and 2 IBDU), and mucosal samples were collected from both ileum and colon from 63 patients before, during, and after IFX treatment. The bacterial microbiota composition was investigated by targeting the conserved 16S region in MiSeq sequencing. Additionally, the relative sequencing data were quantified by qPCR. Responders to IFX had an increase in the total bacterial load in ileum and fecal samples during treatment, primarily driven by butyrate-producing bacteria in the Firmicutes phylum. Interestingly, this was only observed in the fecal samples in responders, but not non-responders to IFX. These results indicate that the gut bacterial microbiota of responders to IFX is changing toward a more favorable composition during successful IFX treatment.

**IMPORTANCE**  The research described in this paper enhances our understanding of how infliximab (IFX) treatment affects the gut mucosal and fecal microbiota in patients with inflammatory bowel disease (IBD). Using 16S sequencing technique and quantification by qPCR, the study revealed that successful treatment with IFX led to an increase in the total bacterial load in both ileal and fecal samples, as well as a shift in bacterial composition toward a more favorable profile with an increase in butyrate-producing bacteria in the fecal samples in responders but not in non-responders to infliximab. This study emphasizes that the gut microbiota plays an important role in the healing process during infliximab treatment in IBD.

**KEYWORDS**  infliximab, inflammatory bowel disease, gut microbiota

Inflammatory bowel diseases (IBD), with the main subtypes Crohn's disease (CD) and ulcerative colitis (UC), are chronic inflammatory diseases of the gastrointestinal tract. The inflammation can manifest throughout the whole gastro-intestinal tract in CD, while the inflammation is located in the colon in UC. In unclassified IBD (IBDU), the distinction between subtypes cannot be made. The pathogenesis of IBD is still unknown, but many associated risk factors have been extensively studied and identified, including genetic predisposition and multiple environmental factors varying from heavy metal

**Peer Reviewer** Jeremiah Faith, Icahn School of Medicine at Mount Sinai, New York, New York, USA

Address correspondence to Eija Nissilä, eija.nissila@helsinki.fi, or Päivi Saavalainen, paivi.saavalainen@helsinki.fi.

T.S. has received speaker fees from Celltrion, Janssen-Cilag, and Takeda and has served as an advisory board member for Abbvie, BMS, Janssen-Cilag, Pfizer, and Takeda. All other authors declare no conflict of interest.

See the funding table on p. 13.

exposure to maternal smoking (1). IBD cannot be cured, but there are many treatment options including conventional treatment of IBD by corticosteroids, 5-aminosalicylic acid, thiopurines, and methotrexate, aimed at achieving mucosal healing (2, 3). In moderate or severe IBD, however, further treatment with biologics is needed. Tumor necrosis factor alpha (TNF-α) blockers such as infliximab (IFX) are commonly used in first-line biological treatment after conventional therapy. IFX is a biological medicine that binds to the proinflammatory mediator TNF-α, thereby blocking it from binding to its receptor, and reducing inflammation effectively. The problem, however, is that up to half of the patients do not have a good long-term response to the drug (4–7).

The gastro-intestinal tract is colonized by microorganisms, including bacteria, fungi, viruses, and archaea, making up the gut microbiota. The gut bacteria are by far the most abundant in the gut microbiota and is composed of the following phyla: Firmicutes and Bacteroidota, which are the most abundant, and Actinobacteria, Proteobacteria, Fusobacteria, and Verrucomicrobia. As previously reviewed (8), the gut microbiota composition is uniquely individual and is affected by genetics, environment, and lifestyle factors such as diet. Although this makes it difficult to specify what a healthy gut microbiota is, it has been widely studied and is characterized with a high diversity, gene richness, and stability in its composition (8). Changes in the gut microbiota shifting it into an unfavorable composition are linked to various diseases. Many chronic inflammatory diseases have been observed to have a strong link to aberrant gut microbiota composition (9), including IBD where an increase in facultative anaerobes and a decrease in obligately anaerobic bacteria, particularly those producing short-chain fatty acids (SCFA) has been observed (10).

The gut microbiota composition differs between the different sites of the gastrointestinal tract, with the large intestine having a higher bacterial count compared to the small intestine. Furthermore, the bacterial count is higher in the ileum compared to the other parts of the small intestine as previously reviewed (11). Fecal samples have been widely used as samples of gut microbiota, and most studies present results based on this. Fecal samples present a sum of the transition through the whole GI tract and are further affected by factors such as transit time and food intake. Previously, it has been shown that fecal samples differ from mucosal samples (12, 13).

The possibility to predict the response to IFX therapy based on the gut microbiota has previously been investigated, and it has been observed that the relative abundance of butyrate-producing bacteria such as bacteria in the Clostridia class is higher in responders compared to non-responders to IFX (14). The bacterial profile before the start of IFX treatment from the fecal data previously published presented that the IFX response could be predicted with high accuracy (15). The mechanisms behind treatment response are still unknown, and as reviewed by Mah et al., the cohorts are heterogeneous and small and the follow-ups are short, making it difficult to draw conclusions (14). Additionally, to our knowledge, there are only two previous studies on the gut mucosal microbiota during IFX treatment (16, 17), but no fecal samples were included in them, and the study design and methods differed from this study. The aim of our study was to investigate the quantitative gut mucosal bacterial microbiota composition during IFX treatment in responders to IFX. Additionally, the aim was to compare these results to the quantitative changes of the fecal bacterial composition during the therapy and to investigate what happens in the quantitative fecal microbiota of non-responders during IFX treatment.

## MATERIALS AND METHODS

### Study design

In this study, 75 adult IBD patients were recruited at the Department of Gastroenterology at Helsinki University Hospital from February 2017 to February 2019, for whom IFX treatment was started due to active inflammation and non-response or intolerance to previous treatments with conventional or biological medication. However, three patients

withdrew consent, and two were excluded due to immunization and surgery. Thereby 70 patients were included in the study (Fig. 1). Biopsy samples were collected when endoscopy was performed before initiating IFX treatment, around week 12, and at 1 year post-treatment initiation from both inflamed and non-inflamed regions (if available) of the terminal ileum and the colon. Biopsies were taken with standard biopsy forceps. Additionally, multiple biopsies were collected from the terminal ileum and different segments of the colon for routine histopathologic assessment of disease activity. The biopsies for the mucosal microbiota analysis were immediately transferred into RNA*later* (ThermoFisher Scientific) and stored at −20°C until extraction. Fecal samples were collected before the start of IFX and at 2, 6, 12 weeks, and 1 year post-treatment initiation as previously described (15). The study design and sample collection are outlined in Fig. 1.

## Evaluation of response

Patients underwent colonoscopy before IFX induction at baseline and around week 12 after treatment initiation. Baseline disease activity and IFX treatment response were assessed with endoscopic indices. In some patients, baseline endoscopic scores were missing, and for these patients, scores needed to be reconstructed retrospectively by a gastroenterologist based on the pictures taken during the colonoscopy. For CD patients, the Simple Endoscopic Score for Crohn's Disease (SES-CD), and for UC patients, the Mayo Endoscopic Subscore (MES) were applied. In CD, endoscopic remission was defined as an SES-CD ≤2 points, and endoscopic response as an SES-CD decrease of ≥50% from baseline (18). In UC, an MES of ≤1 was regarded as remission and a decrease of ≥1 from baseline as a response (19).

In patients with endoscopically unreachable disease locations, e.g., small intestine CD, or patients refusing to undergo colonoscopy, baseline disease activity, and week 12 treatment response were assessed with a combination of a clinical score and fecal calprotectin (FC). For CD patients, the Harvey-Bradshaw-Index (HBI), and for UC patients, the Partial Mayo Score (PMS) were used. In CD, clinical and biochemical remission was defined as an HBI ≤ 4 points combined with FC ≤200 µg/g, and clinical and biochemical response as a decrease in HBI ≥ 3 points combined with an FC decrease of ≥50% from baseline (20, 21). In UC, a PMS ≤2, with no individual subscore >1, combined with an FC ≤200 µg/g, was regarded as clinical and biochemical remission, and a PMS decrease ≥3 points combined with an FC decrease ≥50% from baseline as response (21, 22).

## DNA extraction and MiSeq library preparation

The 16S rRNA gene amplicon data from fecal samples were acquired from the previous study (15). For the biopsy samples, RNA*later* was first removed, and the biopsy was further washed twice with ice-cold PBS to remove all RNA*later*. Ice-cold PBS (200 µL) was added to the biopsy and vortexed for 1 min, followed by 10 min incubation on ice, followed again by a 2 min vortex. The supernatant was used in DNA extraction using a repeated bead-beating method as previously described (23) and was purified using KingFisher Flex-automated purification system (ThermoFisher Scientific).

The sequencing library was prepared first with an enrichment phase due to the low DNA concentrations using the primers REV 5′-AAG GAG GTG ATC CAR CCG CA-3′ and FOR 5′-GAG AGT TTG ATY CTG GCT CAG-3′ (24) targeting the full length of the 16S rRNA sequence (20 cycles) followed by 16S library preparation as previously (15). In the enrichment phase, 5 ng DNA template, 1 µL of each primer (10 µM), 10 µL 2× Phusion High-Fidelity PCR Master Mix with HF Buffer (Thermo Scientific), and 3 µL water in a PCR of 20 µL were used. The following PCR conditions were used: 94°C for 2 min followed by 20 cycles of 94°C for 20 s, 55°C for 20 s, 72°C for 1.5 min, and finally 72°C for 5 min. In the following 16S PCR, 4.25 µL of unpurified PCR product was used as template and 0.75 µL DMSO was added to a PCR of 20 µL. The 16S library was sequenced with Illumina MiSeq

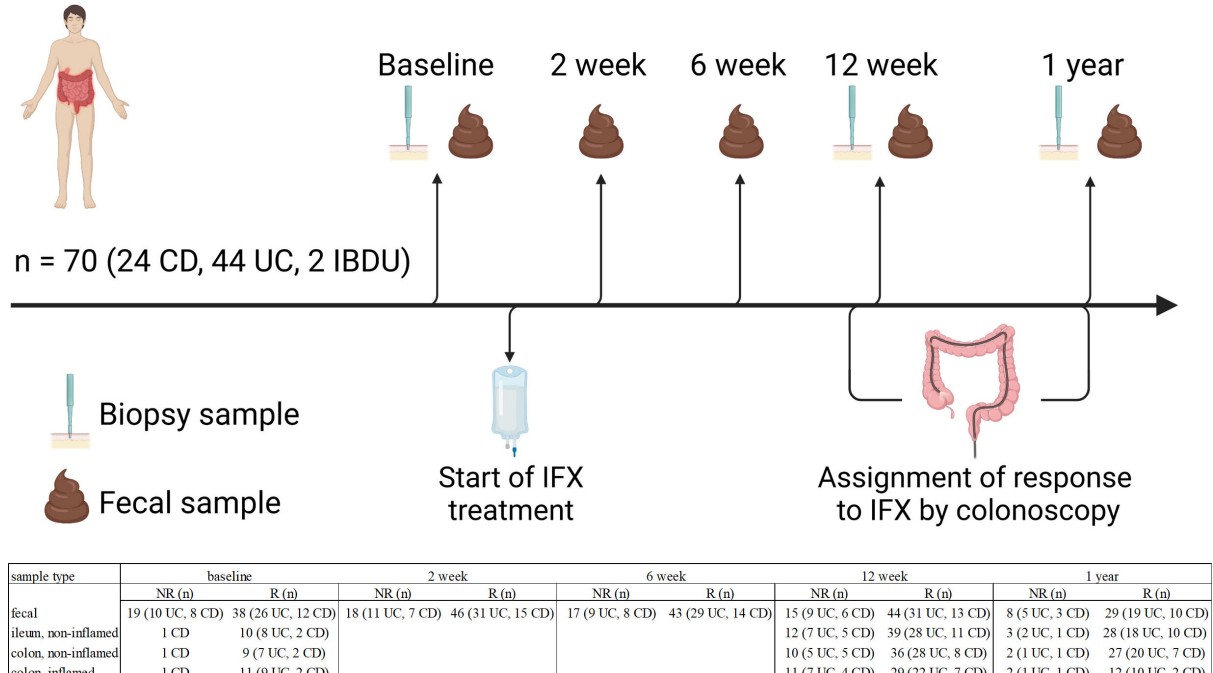

FIG 1 Overview of the sample collection. CD, Crohn's disease; UC, ulcerative colitis; IBDU, unclassified inflammatory bowel disease; IFX, infliximab; R, responder to IFX; NR, non-responder to IFX.

| sample type | baseline | | 2 week | | 6 week | | 12 week | | 1 year | |
|---|---|---|---|---|---|---|---|---|---|---|
| | NR (n) | R (n) | NR (n) | R (n) | NR (n) | R (n) | NR (n) | R (n) | NR (n) | R (n) |
| fecal | 19 (10 UC, 8 CD) | 38 (26 UC, 12 CD) | 18 (11 UC, 7 CD) | 46 (31 UC, 15 CD) | 17 (9 UC, 8 CD) | 43 (29 UC, 14 CD) | 15 (9 UC, 6 CD) | 44 (31 UC, 13 CD) | 8 (5 UC, 3 CD) | 29 (19 UC, 10 CD) |
| ileum, non-inflamed | 1 CD | 10 (8 UC, 2 CD) | | | | | 12 (7 UC, 5 CD) | 39 (28 UC, 11 CD) | 3 (2 UC, 1 CD) | 28 (18 UC, 10 CD) |
| colon, non-inflamed | 1 CD | 9 (7 UC, 2 CD) | | | | | 10 (5 UC, 5 CD) | 36 (28 UC, 8 CD) | 2 (1 UC, 1 CD) | 27 (20 UC, 7 CD) |
| colon, inflamed | 1 CD | 11 (9 UC, 2 CD) | | | | | 11 (7 UC, 4 CD) | 29 (22 UC, 7 CD) | 2 (1 UC, 1 CD) | 12 (10 UC, 2 CD) |

sequencing at the Biomedicum Functional Genomics Unit (FuGU), University of Helsinki, Finland.

The quantification of both the fecal and biopsy MiSeq sequencing data was done by qPCR using the universal bacterial primers 331 F/797 R (25) as previously described (26) using 0.5 ng of fecal DNA or 5 ng of mucosal DNA (Bio-Rad CFX96). The 16S rRNA copy numbers in one biopsy, or per g of fecal matter, were calculated by using a standard of full-length amplicons of *Bifidobacterium bifidum* 16S rRNA gene to convert threshold cycle values (Ct) into copy numbers. The absolute abundances were calculated by multiplying the relative abundances with the total copy numbers.

The 16S rDNA sequences and the copy numbers are available in the European Nucleotide Archive (project PRJEB51062 [sequences from fecal samples] and PRJEB87512 [sequences from biopsy samples]).

## Analysis of sequencing data

Both biopsy and fecal 16S sequencing data were processed with the DADA2 pipeline (27), using the default parameters. In the mucosal 16S data, both forward and reverse reads were included and merged according to the pipeline, but in the fecal samples, only forward reads were used due to the low quality of reverse reads. Reads were annotated to the Silva database (version 138.1) (28). The median number of preprocessed and annotated reads in the mucosal 16S data was 5569. An increase in richness did not correlate with an increase in read count. The median number of preprocessed and annotated reads from the re-processed fecal sequencing data was 25,138 reads.

## Statistical analysis

For all statistical analyses of the absolute 16S sequencing data from both the mucosal samples and fecal samples, the R package mare (29) was used, including tools from packages vegan (30), MASS (31), and nmle (32). All $P$-values from differences between taxonomies were corrected for false discovery rate (FDR) (33). The effect of background and technical factors was analyzed using principal coordinates analysis (PCoA), with R

package vegan (30) and multivariate permutational analysis of variance (PERMANOVA). The effect of factors that were significantly associated with the microbiota composition was eliminated from the results, and these were IBD subtype, smoking, sex, age, and the batch effect from qPCR run and MiSeq run and additionally the medications for the fecal data. For the comparison of copy numbers and visualization of the absolute abundance of taxonomies over time, GraphPad Prism 9.0 (GraphPad Software, Inc., San Diego, CA) was used. Mann – Whitney $U$ test was used in the comparisons.

## RESULTS

### Patient characteristics

Patient characteristics are presented in Table 1. Initially, 75 patients were recruited, 3 withdrew consent and were excluded from the study. Two more patients were excluded before week 12, one due to upper GI surgery and the other due to immunization to IFX. Fecal samples of 70 adult IBD patients starting IFX therapy were, thus, included in the 16S MiSeq sequencing. Biopsies were collected from 62 patients. The number of samples at each time point and location is specified and presented in Fig. 1.

### Response to IFX

The response to IFX was evaluated from 68 patients at week 12 (Fig. 2). Two patients terminated IFX treatment before week 12 due to non-response and were considered non-responders (NRs). At week 12, 47 patients were in remission (Rs) (67.1%), 4 patients showed partial response (PRs) (5.71%) to IFX, and 19 patients NRs (27.1%). Between weeks 12 and 1 year, 15 patients discontinued (12 due to lack of response and were thus considered NRs). At 1 year, 42 patients were Rs (64.6%), 3 PRs (4.6%), and 20 NRs (30.8%) (Fig. 2).

As the response varied between week 12 and 1 year, four different response groupings have been used in the statistical analyses: (i) response groups according to response evaluation that remained the same at both evaluations; (ii) response groups that remained the same at both evaluations, including also those cases where PR at one time point changed to another response group at the other time point; (iii) 12-week response evaluation; and (iv) 1-year response evaluation. The results were then compared between these options and only those results that were significant in multiple tests are presented here. The response from week 12 is used in the figures.

**TABLE 1** Clinical characteristics of patients[a]

| Characteristics | n (%) or Median (min-max) |
|---|---|
| No. of patients | 70 |
| Female | 29 (41.4) |
| Crohn's disease (CD) | 24 (34.3) |
| Ulcerative colitis (UC) | 44 (62.9) |
| IBD unclassified | 2 (2.9) |
| Age at diagnosis, years | 25 (12–52) |
| Age at IFX initiation, years | 32 (12–63) |
| Disease duration, years | 4 (0–40) |
| Previous surgery | 9 (12.9) |
| Smoking | 11 (15.7) |
| Concominant medication at IFX initiation | |
| Steroid | 35 (50.0) |
| 5-Aminosalicylic acid | 37 (52.9) |
| Methotrexate | 4 (5.7) |
| Metronidazole | 5 (7.1) |
| Ciprofloxacin | 6 (8.6) |
| Thiopurine | 49 (70.0) |

[a]IBD, inflammatory bowel disease; IFX, infliximab.

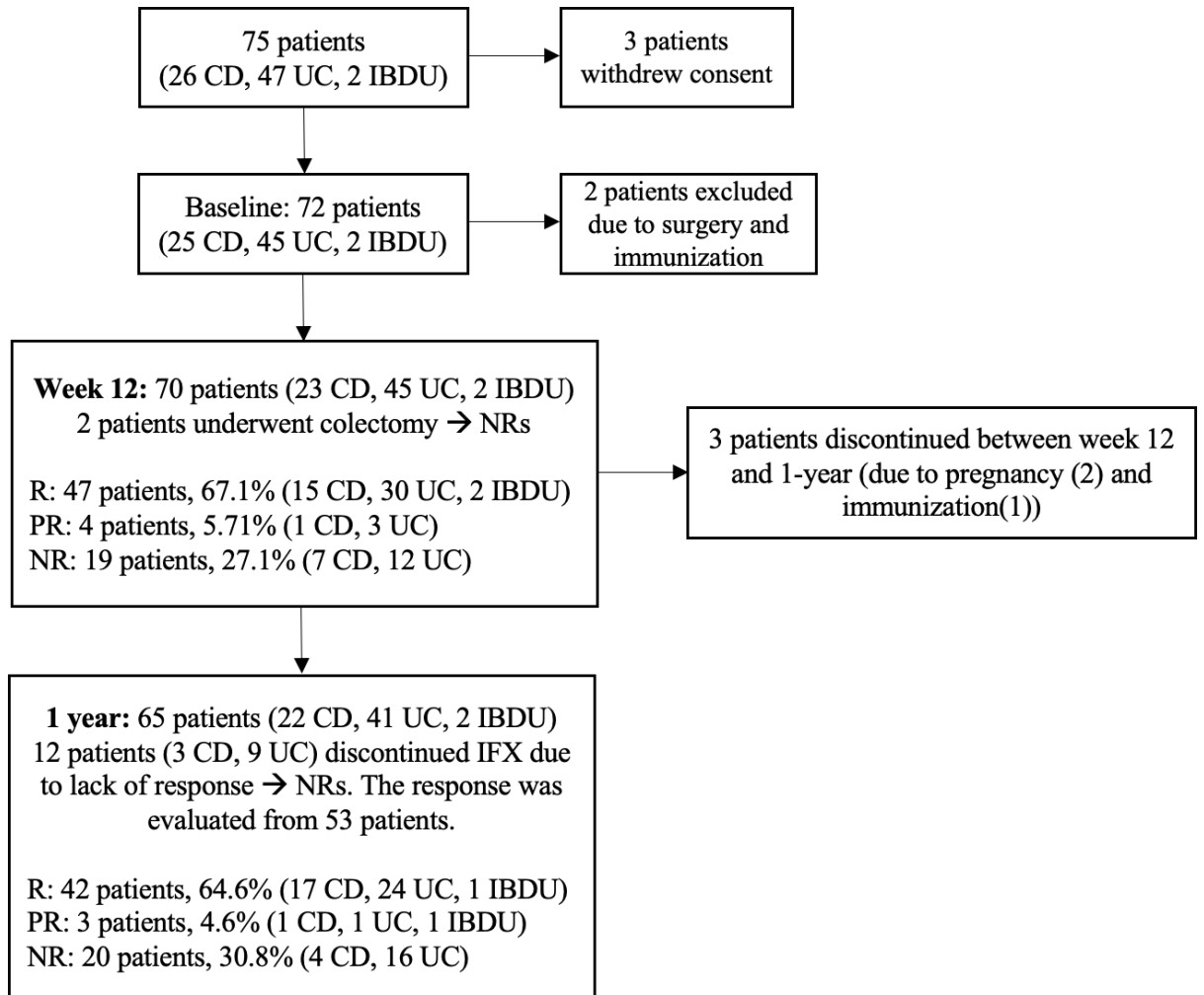

**FIG 2** Number of patients and response evaluation at each time point. CD, Crohn's disease; UC, ulcerative colitis; IBDU, unclassified inflammatory bowel disease; IFX, infliximab; R, responder to IFX, PR, partial responder to IFX, NR, non-responder to IFX.

## Overview of mucosal microbiota composition

The mucosal bacterial microbiota consisted of Firmicutes (47.4%), Bacteroidota (35.0%), and Proteobacteria (14.3%), making up 96.7% of the total bacterial microbiota. Furthermore, 244 genera were represented, of which *Bacteroides*, *Faecalibacterium*, *Escherichia-Shigella*, *Anaerostipes*, *Blautia,* and *Lachnoclostridium* were the most abundant, making up 63.6% of the microbiota, calculated by combining all samples.

## Diversity, richness, and copy number

In the mucosal data, no difference was observed in bacterial diversity or richness between the time points during IFX treatment (Fig. S1). The diversity and richness of the bacterial fecal microbiota composition have previously been published (15), but no difference was observed in either response group over time.

In the non-inflamed ileum samples of responders, the bacterial copy number was significantly higher at 1 year compared to before the start of IFX treatment ($P = 0.0282$; Fig. 3A). The same trend was seen when stratifying according to the IBD subtypes although not significant. This was also observed in the fecal samples, where the copy numbers were significantly higher at week 6 ($P = 0.032$), week 12 ($P = 0.0316$), and 1 year ($P = 0.0386$) post-IFX compared to baseline in Rs (Fig. 3B). When analyzing the copy numbers per gram of feces of CD patients, no significant difference was observed

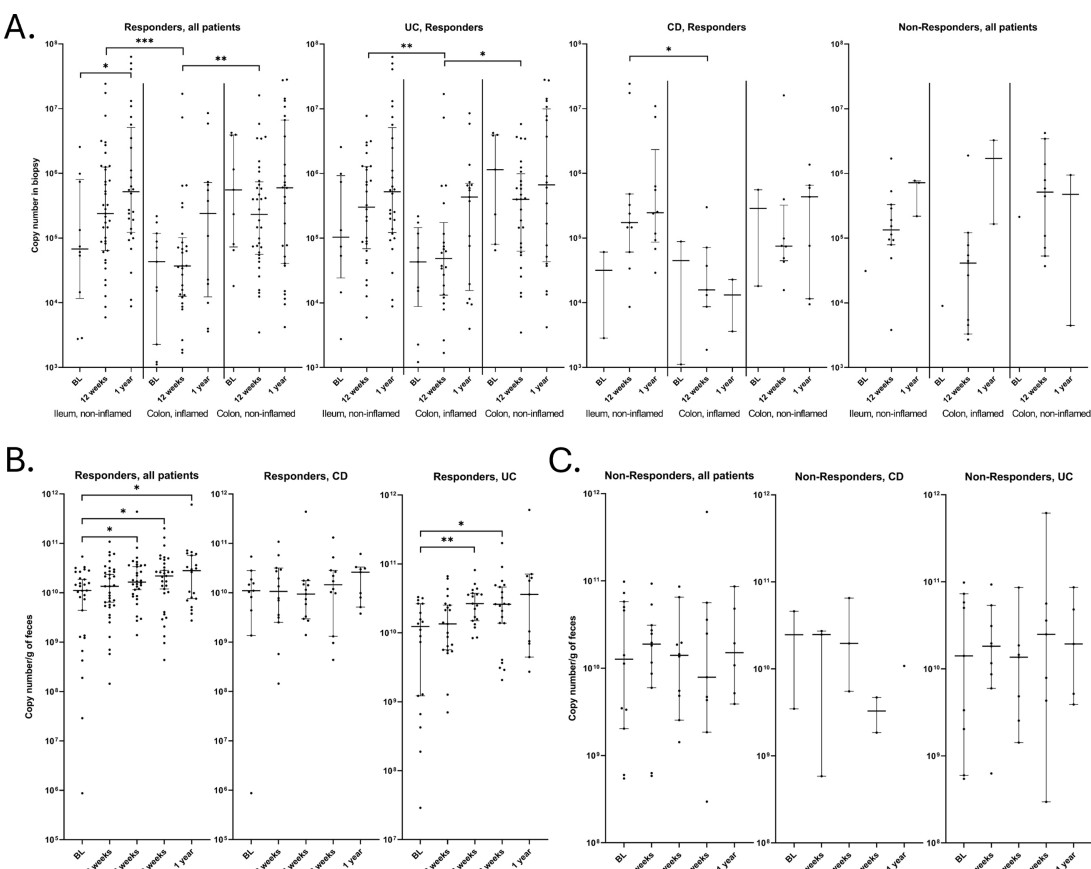

**FIG 3** The total bacterial copy number before start of infliximab treatment (BL), and week 12, and 1 year post treatment stratified by (A) mucosal samples of responders to IFX, (B) fecal samples of responders, and (C) fecal samples of non-responders to infliximab. BL = baseline, before start of infliximab treatment, CD = Crohn's disease, UC = ulcerative colitis.

although the bacterial load increased in the Rs. In UC/IBDU patients, the fecal copy numbers were significantly higher at week 6 ($P = 0.0021$) and week 12 ($P = 0.0221$) compared to baseline in Rs (Fig. 3B). This trend was not detected in NRs (Fig. 3C). The copy numbers were further significantly elevated in both non-inflamed ileum sample and non-inflamed colon sample compared to inflamed colon sample (Fig. 3A).

## Gut mucosal microbiota in non-inflamed ileum samples of responders to IFX during treatment

When comparing the gut microbiota composition at baseline to week 12 and 1 year in non-inflamed ileum samples of Rs to IFX (Fig. 4), there were significant differences at all taxonomical levels (Fig. S2A through E). When stratifying by IBD subtype, the same taxa were significantly increased in UC/IBDU patients, except for the class Alphaproteobacteria, the order Propionibacteriales, the family *Prevotellaceae* (significant but negative FC), and genera *Blautia*, *Agathobacter*, *Fusicatenibacter*, *Roseburia*, and *Dialister* (significant, but negative FC). Additionally, the phyla Actinobacteriota, the classes Actinobacteria and Negativicutes, the family Clostridiales, orders *Bacteroidaceae* and *Clostridiceae,* and genera *Bradyhizobium* and *Bacteroides* increased significantly during IFX treatment, while the genus *Eubacterium eligens* group was more abundant at baseline.

In CD, however, the results were less clear, and additionally, statistically significant differences were possible to calculate only between 1 year and baseline due to the low number of samples. It was observed that the family *Bacteroidaceae* and genus *Bacteroides* decreased significantly over time, in contrast to UC/IBDU patients. Furthermore, the order Burkholderiales, its family *Sutterellaceae* and genus *Sutterella*, the order

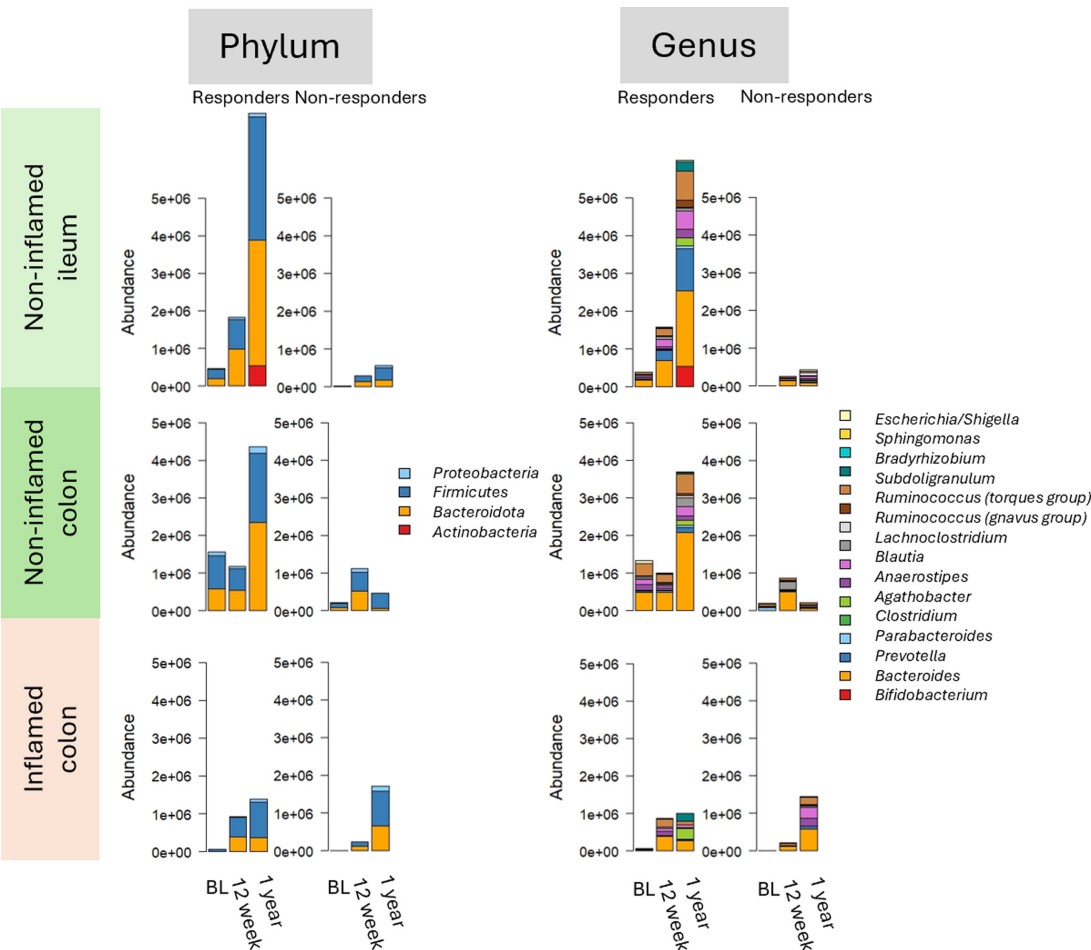

**FIG 4** The mucosal microbiota composition presented at phylum and genus level during infliximab therapy in non-inflamed ileum, non-inflamed colon, and inflamed colon samples of responders and non-responders to infliximab (including all patients). The bacterial taxonomies are color-coded and presented on the right side of the plots. BL = baseline, before start of infliximab treatment.

Enterobacteriales, its family *Enterobacteriaceae,* and the genus *Escherichia-Shigella* group (all belonging to the class Gammaproteobacteria) were more abundant at baseline compared to later time points. Additionally, *Prevotellaceae* increased during therapy, and furthermore, the order Clostridiales, its family *Clostridiaceae*, and genera *Clostridium* (*sensu stricto* group), *Ruminococcus* (gnavus group), and *Agathobacter* (all belonging to the class Clostridia) increased during therapy. Finally, the phylum Proteobacteria, class Alphaproteobacteria, order Veillonellales-Selenomonadales group, and its family *Veillonellaceae* also increased during IFX therapy.

Additionally, when looking at the relative abundance (% of the absolute abundance), the Bacteroidota, Bacteroidia, and Bacteroidales decreased during therapy in all patients.

## Gut mucosal microbiota in colon samples of responders to IFX during treatment

The results were less consistent in the colon samples compared to ileum samples, indicating a bigger variety. The variation was even greater in the inflamed compared to the non-inflamed samples. It was not possible to study CD patients separately due to the low number of colon samples. There were significant differences in the non-inflamed samples when comparing weeks 2 and 6 to baseline (Fig. 4) at all taxonomical levels of Rs to IFX (Fig. S3A through E). Here, the family *Prevotellaceae* was more abundant before

the start of treatment although the prevalence was low (Fig. S3D). All other taxa were significantly increased after treatment.

In the inflamed samples of Rs to IFX, the order Veillonellales-Selenomonadales group and its family *Veillonellaceae* increased, while the class Alphaproteobacteria and its order Rhizobiales decreased during treatment (Fig. S4). When analyzing the UC/IBDU group separately, no taxa increased or decreased during treatment.

### Fecal microbiota during IFX treatment

When studying the fecal microbiota, the Rs had more taxa that consistently increased during treatment (Fig. 5). NRs presented more variety and fewer bacterial taxa that were increasing or decreasing over the therapy course. In Rs, significant differences were observed at all taxonomical levels in UC/IBDU and at some levels in CD patients, as outlined in Fig. 5 (specifics in Fig. S5 and S6). When combining both CD and UC/IBDU patients, there were no clear results. The results were more consistent in UC/IBDU patients compared to CD patients. In UC/IBDU patients, the abundance of the genus *Megasphaera* was significantly lower at other time points compared to baseline although the abundance was low. Other taxa increased over time (Fig. S5A through E).

All taxa that differed significantly during treatment in CD patients are presented (Fig. S6A through D). The genus *Eubacterium ruminantium* group was significantly more abundant at baseline, with a low abundance. Other taxa increased significantly over time. One order, three families, and three genera were more abundant in UC/IBDU NRs at baseline compared to during treatment (Fig. S6E through G). The abundances are, however, low. The family *Coriobacteriaceae* and genus *Collinsella* increased during treatment in CD NRs, as also reported in CD Rs (Fig. S6C and D), but the abundances are low in NRs.

### Difference in mucosal microbiota between IFX response groups

The difference in bacterial mucosal microbiota between response groups was possible to study only at week 12 post-IFX treatment due to the lack of samples at other time points. There was no difference in diversity, richness, or copy number. Multiple bacterial taxa differed significantly between response groups in non-inflamed ileum (Table S1), colon (Table S2), and inflamed colon (Table S3). The phylum Actinobacteriota, class Actinobacteria, and genus *Agathobacter* were significantly ($P < 0.05$) more abundant in Rs compared to NRs in all three biopsy groups. *Dorea* was more abundant in Rs compared to NRs in non-inflamed ileum and colon, but more abundant in NRs compared to Rs in inflamed colon samples.

### DISCUSSION

Here, we have studied the quantitative composition of the bacterial gut microbiota from non-inflamed ileum, non-inflamed colon, inflamed colon, and fecal samples during IFX treatment in adult IBD patients. The relative sequencing data were analyzed by MiSeq sequencing and quantified by qPCR to acquire the absolute abundances. The differences in the bacterial gut microbiota between the IFX response groups from the fecal samples included here have previously been published (15). However, to our knowledge, this is the first study of the quantitative bacterial composition during IFX treatment in both mucosal and fecal samples. Mucosal microbiota might give more insights into the effects of IFX treatment as it reflects a more local, direct interaction with the immune system compared to only studying fecal microbiota. Quantitative bacterial composition during IFX treatment has previously been studied in mucosal samples (16), although not including fecal samples, and only rectum biopsies. Additionally, the sequencing method was different. Here, we have observed that the total bacterial load increased in responders, but not in non-responders, during treatment, driven particularly by butyrate-producing bacteria belonging to the Firmicutes, indicating a more favorable microbiota composition.

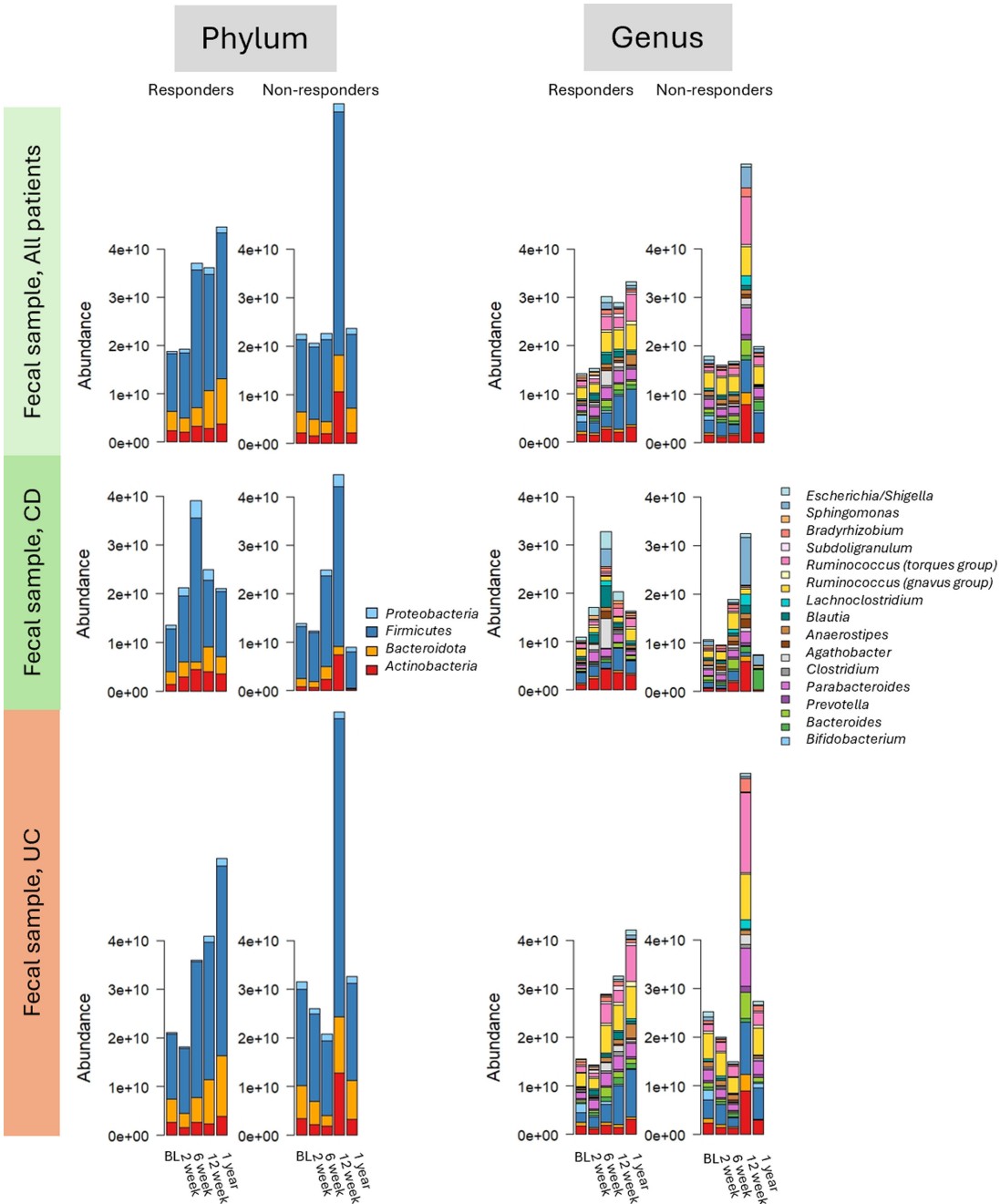

**FIG 5** The fecal microbiota composition presented at phylum and genus level during infliximab therapy stratified by subtype and response to infliximab. The bacterial taxonomies are color-coded and presented on the right side of the plots. BL = baseline, before start of infliximab treatment, CD = Crohn's disease, UC = ulcerative colitis.

The non-inflamed ileum, non-inflamed colon, and inflamed colon were studied in Rs to IFX. The results differed between the sample types although a previous study presented no differences (34). The bacterial load was lower in the inflamed colon compared to the non-inflamed ileum and colon samples. In the inflammatory state of the gut, the mucosal layer becomes thinner, indicating that the mucosal layer of non-inflamed tissue is thicker than the inflamed (17). This could explain why the bacterial load was lower in the inflamed colon samples compared to the other groups. Previous studies have shown that the bacterial load is higher in the large intestine compared to the small (11). Here, we did not observe such differences. However, a recent review

also indicated that there is a separate bacterial community colonizing the area from duodenum to proximal ileum, and another from distal ileum to rectum (35). As the samples included here are from either the ileum or colon, it is possible that the difference is not so significant. The specific location of the colon sample did vary based on the location of the inflammation, which might also impact the observed differences. Overall, the non-inflamed ileum samples presented more consistent results, while there was more variation among the colon samples, as expected—additionally, the inflamed colon samples presented even less consistent results. Inflamed colon biopsies were collected from all patients showing even some indication of endoscopic inflammation, so they also include patients classified as Rs to IFX according to the definition of endoscopic remission (MES < 1, SES-CD ≤ 2), making this group heterogeneous. This could explain the larger variation in the results of this group as well. Furthermore, the CD patients presented more variety in the results compared to UC/IBDU, which is also expected as the disease location and type are more variable in CD compared to UC, where the disease is more specifically located in a separate part of the colon. Due to the low number of samples, the mucosal microbiota of NRs was not studied during IFX therapy.

The gut microbiota has been studied during IFX therapy previously (14). However, the methods vary between studies, and very few have investigated the absolute abundance of bacteria, particularly the mucosal microbiota. Studying only the relative abundance can give false insights, particularly when analyzing the changes in the gut microbiota over time (26). Additionally, the responders and non-responders are often not analyzed separately. Here, we have presented clear differences between the response groups in the fecal samples, and we have found many correlations between our results and previously published results.

We found that the total number of bacteria increased in fecal and non-inflamed ileum samples of Rs during IFX treatment, but this was not observed in NRs. The increase in bacterial load here might reflect the mucosal healing. As the gut heals, symptoms such as diarrhea and blood in stool are also alleviated, which, in turn, affects both the mucosa and the fecal composition. Additionally, this might give an opportunity for bacterial growth as well. It is established that the total bacterial load of the gut microbiota is higher in healthy controls compared to patients with IBD (36, 37) that further strengthens that the increase in bacterial load observed here indicates a healthier state of the gut microbiota composition. This increase in the total bacterial load was driven by bacteria belonging to the Firmicutes, and its class Clostridia, which increased in Rs during IFX treatment, as observed in non-inflamed ileum and non-inflamed colon samples of all patients and fecal samples of UC/IBDU patients. These were also more abundant in non-inflamed ileum samples of Rs compared to NRs at 12 weeks post IFX. Furthermore, the order Clostridiales and its family *Lachnospiraceae* increased in both CD and UC patients in the non-inflamed ileum samples, and many of its genera were also observed to be increased in the different groups. Clostridia has previously been observed elevated in Rs to IFX already before start of IFX (14), also in the fecal samples included here (15), and in our recent study on pediatric patients with IBD as well (38). Additionally, previous studies have observed an increased relative abundance of Clostridiales (39) and its order *Lachnospiraceae* (40) during IFX treatment.

It has also been reported that the relative abundance of the class Bacteroidia decreased during IFX treatment (41), and further the microbiota composition changed from Bacteroidota to Firmicutes (42). Here, the absolute abundance of Bacteroidia increased in many sample types, except in the non-inflamed ileum samples of CD patients where it decreased. Interestingly, the relative abundance decreased in non-inflamed ileum, highlighting the importance of focusing on quantified sequencing data when analyzing changes in the microbiota over multiple time points. In IBD, the reduction in diversity has previously been associated with a decrease of Firmicutes, particularly *Faecalibacterium prausnitzii* (43), being replaced by unfavorable bacteria, particularly Bacteroidetes and the family *Enterobacteriaceae* (44, 45). Firmicutes are associated with anti-inflammatory properties, and many of their species are

butyrate-producers. Butyrate is a short-chain fatty acid produced by bacteria in the gut microbiota, associated with many anti-inflammatory properties, that has also been observed to be less expressed in NRs to TNF-α treatment (46).

Furthermore, an increase in the relative abundance of *Blautia* (40), *Roseburia* (17, 40), *Ruminococcus* (17), *Anaerostipes* (17), *Clostridium* (47) (belonging to the class Clostridia), *Akkermansia* (40) *Collinsella* (47), and absolute abundance of *Veillonella*, *Ruminococcus,* and *Parabacteroides* (16) during IFX treatment has previously been reported. Additionally, a decreased relative abundance of Enterobacteriales, its family *Enterococcaceae* (48*)*, and its genus *Escherichia-Shigella* group and *Enterobacter* (40) were reported in Rs to IFX. These results agree with the results presented here. We observed a higher absolute abundance of *Enterococcaceae* and its genus *Enterococcus* before the start of IFX in NRs in UC/IBDU fecal samples, and Enterobacteriales, its family *Enterobacteriaceae,* and genus *Escherichia-Shigella* decreased in non-inflamed ileum samples from CD Rs. However, the abundances in the fecal samples of NRs were very low, making it difficult to state an actual difference.

The strengths of this study are the rapid freezing of the fecal samples, longitudinal sampling, and the response evaluation based on endoscopic scores, making them more reliable. Also, the quantification of all sequencing data enabled comparison between time points. The limited number of patients is acknowledged as a limitation, particularly CD patients, in addition to the limited number of mucosal samples and lack of a healthy control group. Finally, the lack of information on diet is a limitation as it is known to impact the gut microbiota composition as well.

In a recent study, it was confirmed that the changes in the gut bacterial microbiota of IFX-treated patients were due to mucosal healing, and not to the medication (42). It is still unknown whether the successful treatment could be partly due to responders having a more favorable gut microbiota composition before the start of treatment, or if the gut microbiota reflects of the overall health or state of the immune system in the patient. This study, together with previously published studies, indicates that the gut microbiota might play an important role in the treatment outcome of IFX therapy.

In conclusion, this study provides insights into the dynamics of the gut microbiota during IFX treatment in IBD patients. It highlights the importance of quantitative analysis and suggests a link between microbiota changes, mucosal healing, and treatment response. Understanding these relationships could help improve IBD management and tailor treatments for individual patients.

## Lay summary

Successful treatment with IFX in IBD leads to an increase in the total bacterial load in both ileum and fecal samples, as well as a shift in bacterial composition towards a more favorable composition characterized by an increase in butyrate-producing bacteria.

## ACKNOWLEDGMENTS

We would like to acknowledge the excellent assistance by study nurses Pirkko Tuukkala and Virpi Pelkonen. Additionally, we thank Hanne Ahola and Katarzyna Leskinen for handling the samples, Tinja Kanerva and Pinja Elomaa for assisting in DNA extraction, Anne Salonen for help and support in the choosing of methods, and Heli Pessa for the thorough language review.

## AUTHOR AFFILIATIONS

[1]Human Microbiome Research Program, Faculty of Medicine, University of Helsinki, Helsinki, Finland
[2]Folkhälsan Research Center, Helsinki, Finland
[3]Department of Gastroenterology, Helsinki University Hospital and University of Helsinki, Helsinki, Finland

[4]Department of Bacteriology and Immunology, Medicum, University of Helsinki, Helsinki, Finland

[5]University of Helsinki, Translational Immunology Research Program, Helsinki, Finland

## AUTHOR ORCIDs

Rebecka Ventin-Holmberg ⓘ http://orcid.org/0000-0001-9890-1066
Eija Nissilä ⓘ http://orcid.org/0000-0002-8089-1266
Päivi Saavalainen ⓘ http://orcid.org/0000-0002-1399-1689

## FUNDING

| Funder | Grant(s) | Author(s) |
|---|---|---|
| Mary och Georg C. Ehrnrooths Stiftelse | | Rebecka Ventin-Holmberg |
| Orionin Tutkimussäätiö | | Rebecka Ventin-Holmberg |
| Victoriastiftelsen | | Rebecka Ventin-Holmberg |
| Biomedicum Helsinki-säätiö | | Rebecka Ventin-Holmberg |
| HY \| Folkhälsan Research Center, University of Helsinki (Folkhälsan Research Center) | | Julia Eriksson |
| | | Päivi Saavalainen |
| University of Helsinki | | Rebecka Ventin-Holmberg |

## AUTHOR CONTRIBUTIONS

Rebecka Ventin-Holmberg, Data curation, Formal analysis, Investigation, Methodology, Software, Validation, Visualization, Writing – original draft, Writing – review and editing | Julia Eriksson, Data curation, Investigation, Methodology, Writing – review and editing | Anja Eberl, Conceptualization, Data curation, Formal analysis, Investigation, Project administration, Resources, Writing – review and editing | Taina Sipponen, Conceptualization, Data curation, Funding acquisition, Project administration, Resources, Supervision, Writing – review and editing | Eija Nissilä, Conceptualization, Data curation, Investigation, Methodology, Supervision, Writing – review and editing | Päivi Saavalainen, Conceptualization, Data curation, Funding acquisition, Project administration, Resources, Supervision, Writing – review and editing

## DIRECT CONTRIBUTION

R.V.H., A.E., T.S., E.N., and P.S.: design of the study. R.V.H. and J.E.: extraction of DNA from biopsy, qPCR for biopsy, and fecal samples. R.V.H.: MiSeq library preparation, processing of data, sequencing data analysis and interpretation of results, drafting the manuscript. A.E. and T.S.: recruitment and examination of patients, obtaining biopsy specimens, and analysis of clinical results. All authors revised and accepted the final manuscript.

## ETHICS APPROVAL

The patients recruited in this study signed an informed consent form before starting IFX treatment. The study was approved by the Ethics Committee of the Helsinki University Central Hospital (147/13/03/01/16) and registered in the European Union Drug Regulating Authorities Clinical Trials Database (EUDRA-CT-Number: 2016-001278-13).

## ADDITIONAL FILES

The following material is available online.

### Supplemental Material

**Figure S1 (Spectrum01894-24-s0001.tif).** Diversity and richness in mucosal microbiota.

**Figure S2 (Spectrum01894-24-s0002.tif).** Absolute abundance of bacterial taxonomies in non-inflamed ileum.

**Figure S3 (Spectrum01894-24-s0003.tif).** Absolute abundance of bacterial taxonomies in non-inflamed colon.

**Figure S4 (Spectrum01894-24-s0004.tif).** Absolute abundance of bacterial taxonomies in inflamed colon.

**Figure S5 (Spectrum01894-24-s0005.tif).** Absolute abundance of bacterial taxonomies in fecal samples of Crohn's disease patients.

**Figure S6 (Spectrum01894-24-s0006.tif).** The absolute abundance of bacterial taxonomies that significantly differed in fecal samples of Crohn's disease responders to infliximab and in fecal samples of ulcerative colitis non-responders to infliximab.

**Supplemental legends (Spectrum01894-24-s0007.docx).** Legends for Fig. S1 to S6.

**Table S1 (Spectrum01894-24-s0008.xlsx).** Significant differences in bacterial taxonomies between responders and non-responders to IFX in non-inflamed ileum samples.

**Table S2 (Spectrum01894-24-s0009.xlsx).** Significant differences in bacterial taxonomies between responders and non-responders to IFX in non-inflamed colon samples.

**Table S3 (Spectrum01894-24-s0010.xlsx).** Significant differences in bacterial taxonomies between responders and non-responders to IFX in inflamed colon samples.

## Open Peer Review

**PEER REVIEW HISTORY (review-history.pdf).** An accounting of the reviewer comments and feedback.

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
