## [Reviewer comments · Microbiology Spectrum]

Microbiology Spectrum

The total gut mucosal and fecal bacterial load increases in successful treatment of inflammatory bowel disease with infliximab

Rebecka Ventin-Holmberg, Julia Eriksson, Anja Eberl, Taina Sipponen, Eija Nissilä, and Päivi Saavalainen

Corresponding Author(s): Rebecka Ventin-Holmberg, Helsingin yliopisto - Meilahden kampus

Review Timeline:

Submission Date:	July 29, 2024
Editorial Decision:	March 6, 2025
Revision Received:	April 23, 2025
Accepted:	May 13, 2025

Editor: Hao-Yu Liu

Reviewer(s): Disclosure of reviewer identity is with reference to reviewer comments included in decision letter(s). The following individuals involved in review of your submission have agreed to reveal their identity: Jeremiah Faith (Reviewer #2)

Transaction Report:

DOI: <https://doi.org/10.1128/spectrum.01894-24>

Re: Spectrum01894-24 (**The total gut mucosal and fecal bacterial load increases in successful treatment of inflammatory bowel disease with infliximab**)

Dear Dr. Rebecka Venti-Holmberg:

Thank you for the privilege of reviewing your work. Below you will find my comments, instructions from the Spectrum editorial office, and the reviewer comments.

Revision Guidelines

Sincerely,
Hao-Yu Liu
Editor
Microbiology Spectrum

Reviewer #1 (Comments for the Author):

This study investigates the microbiological differences in the ileum and colon biopsies, as well as fecal samples from IBD patients before and after IFX treatment at different time points. The use of absolute abundance further clarifies the microbial data. The types of samples, follow-up period, and the use of absolute abundance provide significant scientific value to this study. However, there are some issues that need to be addressed before publication, particularly regarding the presentation of

data:

1. Presentation of Clinical Data: Although the authors have indicated that the clinical data were previously published, due to the involvement of different groups, it is essential to present the basic information of patients in each group and the results of disease-specific tests in this study. This will help the readers understand the group-specific clinical details.
2. Microbial Analysis: It is generally recommended to include diversity measures (alpha, beta diversity), microbial abundance (grouped stacked bar plots), differential microbes, functional predictions (for 16S, typically PICRUSt2), co-occurrence networks, etc., to provide a comprehensive overview of the microbiota across different groups. Currently, the authors seem to only present comparisons of absolute microbial abundance and diversity that show significant differences, and the figures lack distinct colors and readability. Furthermore, most of the figures present microbial abundance for different groups at each time point. Perhaps the authors could consider displaying the time points on the x-axis only once and combine data from two or three groups at a single time point for comparison (Figures 3, 5, and 6)? Also, considering the redundancy in the x-axis and y-axis information across panels, would it be possible to merge individual small panels into a single figure for better clarity? Ultimately, the figure presentation format is up to the authors, but the current redundancy and lack of readability will be challenging for readers to interpret.
3. Data Accessibility: The authors mention that "The raw 16S rDNA sequences and the copy numbers are available in the European Nucleotide Archive (ENA)." A citation or accession number should be provided for these data.
4. Figure 1: The sample sizes should be adjusted. A tabular format may be more effective in displaying this information, especially for the 2-week and 6-week time points, to better align with other dates. Additionally, words of each sample types such as "small intestine non-inflamed" should be presented in a single line, or other more distinct way.

Reviewer #2 (Comments for the Author):

Reviewer name: Jeremiah Faith (please retain with review)

The manuscript by Ventin-Holmberg et al., provides an interesting look at the bacterial load and absolute microbiome levels of various taxa in the context of people that do or do not respond to infliximab. The cohort is a reasonable size, the conclusions are interesting and novel, and I think the manuscript will be of interest and use to the field. I have only one major comment and several minor comments.

Major comment

1. Similar bacterial loads between colon and small intestine doesn't make a lot of sense. I think it is pretty well established that the loads differ by orders of magnitude between these sites which calls into question the validity of the qPCR assay used or the methodology. Having a more in-depth exploration of this issue would be useful.

Minor comments

1. I really like the focus on absolute quantities and the results are compelling. Working on ileum, colon, and stool in same study is very nice.
2. The manuscript could be better set in the context of the broader field. Several studies have demonstrated a difference in the absolute levels of microbes in IBD vs healthy (Frank et al., 2007, Vandeputte D, et al., Nature 2017, Contijoch et al., eLife 2019) and at least one study has shown responders to therapy (FMT) have higher density (Britton et al., PNAS 2020), aligning with this manuscript but with a different therapy. As a personal conflict of interest, two of these manuscripts are from my own lab. You don't need to cite our papers, but it would be helpful to cite some papers in the field to help establish what already exists and to motivate why look at absolute levels of microbes in IBD, so this one does not feel unnecessarily like an island.
3. Related to the above comment, there is no healthy control comparator group. Although, I think it is too large of an ask to require such a thing in this manuscript, if you do have healthy samples equivalent to these that could be prepared with the same method, it would aid considerably in the interpretation of the results as we cannot conclude if subjects are resolving to normal levels or still below normal levels with the current dataset. If no such data are available, it would be worth mentioning as a minor limitation in the discussion.
4. Lack of diversity in the analyses and plot types. Although I find the results interesting, there is heavy figure redundancy fatigue by the end of this manuscript as all the analyses are variants of the same question using the exact same plot type. Finding different ways to visualize the data would make the paper more readable and could likely yield further insights.
5. When writing results that have the reader compare across plots, it would be helpful to keep the same y-axis across the set of plots. One example is pane 3C, which would be easier to see the key results if all Y axes were 10^7 - 10^{12} .
6. Y axis labels should be more descriptive, so it is easy to know what the sample is and what the units are (copy number / gram stool). Right now everything just says copy number, which is not very helpful.

Response to the reviewer:

Reviewer #1 (Comments for the Author):

This study investigates the microbiological differences in the ileum and colon biopsies, as well as fecal samples from IBD patients before and after IFX treatment at different time points. The use of absolute abundance further clarifies the microbial data. The types of samples, follow-up period, and the use of absolute abundance provide significant scientific value to this study. However, there are some issues that need to be addressed before publication, particularly regarding the presentation of data:

Response: We want to thank the reviewer for the thorough review and insightful comments. Based on your comments we now present an improved and clearer revised version of our manuscript!

1. Presentation of Clinical Data: Although the authors have indicated that the clinical data were previously published, due to the involvement of different groups, it is essential to present the basic information of patients in each group and the results of disease-specific tests in this study. This will help the readers understand the group-specific clinical details.

We agree that a table presenting the basic characteristics of the patients will give a clearer overview for the reader, and the patient characteristics are now added to Table 1 in the revised manuscript.

2. Microbial Analysis: It is generally recommended to include diversity measures (alpha, beta diversity), microbial abundance (grouped stacked bar plots), differential microbes, functional predictions (for 16S, typically PICRUSt2), co-occurrence networks, etc., to provide a comprehensive overview of the microbiota across different groups. Currently, the authors seem to only present comparisons of absolute microbial abundance and diversity that show significant differences, and the figures lack distinct colors and readability. Furthermore, most of the figures present microbial abundance for different groups at each time point.

Perhaps the authors could consider displaying the time points on the x-axis only once and combine data from two or three groups at a single time point for comparison (Figures 3, 5, and 6)? Also, considering the redundancy in the x-axis and y-axis information across panels, would it be possible to merge individual small panels into a single figure for better clarity? Ultimately, the figure presentation format is up to the authors, but the current redundancy and lack of readability will be challenging for readers to interpret.

Thank you for the comment, we agree that the presentation of the results could be improved. In the revised manuscript the Figures 4-7 are moved to the supplements as we find that the presentation of the specific results should be available for the interested, as these provide more in depth overview of the results. We additionally did some changes to improve the readability of Figure 3.

In the revised manuscript we have now included two figures presenting the phylum and genus taxonomical levels and combining the non-inflamed ileum, non-inflamed colon, inflamed colon and fecal samples of responders and non-responders at the different timepoints. In these figures the overall microbial abundances are also presented throughout the study by stacked bar plots to get more basic characteristics of the microbiota composition, as also suggested here. Finally, the diversity and richness measures were added to Supplementary Figure 1, depicted by violin plots.

3. Data Accessibility: The authors mention that "The raw 16S rDNA sequences and the copy numbers are available in the European Nucleotide Archive (ENA)." A citation or accession number should be provided for these data.

The accession numbers are now included in the revised manuscript (P. 8).

4. Figure 1: The sample sizes should be adjusted. A tabular format may be more effective in displaying this information, especially for the 2-week and 6-week time points, to better align with other dates. Additionally, words of each sample types such as "small intestine non-inflamed" should be presented in a single line, or other more distinct way.

We thank you for the insightful comment and have now adjusted the figure according to your suggestions. We, however, decided to keep the graphical figure as we found it helpful for the reader, but have added the number of patients in a tabular format, as suggested here.

Reviewer #2 (Comments for the Author):

Reviewer name: Jeremiah Faith (please retain with review)

The manuscript by Ventin-Holmberg et.al., provides an interesting look at the bacterial load and absolute microbiome levels of various taxa in the context of people that do or do not respond to infliximab. The cohort is a reasonable size, the conclusions are interesting and novel, and I think the manuscript will be of interest and use to the field. I have only one major comment and several minor comments.

Response: We want to thank the reviewer for the thorough review and insightful comments. Based on your comments we now present an improved and clearer revised version of our manuscript!

Major comment

1. Similar bacterial loads between colon and small intestine doesn't make a lot of sense. I think it is pretty well established that the loads differ by orders of magnitude between these sites which calls into question the validity of the qPCR assay used or the methodology. Having a more in-depth exploration of this issue would be useful.

We agree that this was surprising. First, as we do not have access to healthy controls here, we only have a presentation of mucosal tissue in IBD. Although we have the information regarding the inflammatory status of the particular biopsy piece, the whole mucosa is of course largely affected by the ongoing inflammation caused by IBD. Our results that the inflamed colon samples had lower bacterial load compared to the non-inflamed strengthens this hypothesis as well. Additionally, the location of the large intestine sample varied depending on the location of inflammation which also reflects the mucosal microbiota as well. The qPCR assay is a modified protocol from Nadkarni et al. 2002 and has been used for quantification of 16S sequencing data from different sample types as well, highlighting its credibility.

Minor comments

1. I really like the focus on absolute quantities and the results are compelling. Working on ileum, colon, and stool in same study is very nice.

2. The manuscript could be better set in the context of the broader field. Several studies have demonstrated a difference in the absolute levels of microbes in IBD vs healthy (Frank et.al., 2007, Vandeputte D, et.al., Nature 2017, Contijoch et.al., eLife 2019) and at least one study has shown responders to therapy (FMT) have higher density (Britton et.al., PNAS 2020), aligning with this manuscript but with a different therapy. As a personal conflict of interest, two of these manuscripts are from my own lab. You don't need to cite our papers, but it would be helpful to cite some papers in the field to help establish what already exists and to motivate why look at absolute levels of microbes in IBD, so this one does not feel unnecessarily like an island.

As the reviewer pointed out, we have focused primarily on the gut microbiota during infliximab treatment, and therefore already published data is scarce. We, however, agree that including this to the discussion of our results will get a better understanding of the results. We have now cited Contijoch et al. 2019 and Frank et al., 2007 as suggested above.

3. Related to the above comment, there is no healthy control comparator group. Although, I think it is too large of an ask to require such a thing in this manuscript, if you do have healthy samples equivalent to these that could be prepared with the same method, it would aid considerably in the interpretation of the results as we cannot conclude if subjects are resolving to normal levels or still below normal levels with the current dataset. If no such data are available, it would be worth mentioning as a minor limitation in the discussion.

We agree that this is a limitation of our study, but unfortunately, we do not have access to a healthy control group. Therefore, we have according to your suggestion, added this as a limitation in the discussion part of the revised manuscript (P.17). We have additionally not included any statement of subjects returning to healthy levels, as this of course is not possible here. We have instead focused on comparing the results to the response evaluated by medical doctors.

4. Lack of diversity in the analyses and plot types. Although I find the results interesting, there is heavy figure redundancy fatigue by the end of this manuscript as all the analyses are variants of the same question using the exact same plot type. Finding different ways to visualize the data would make the paper more readable and could likely yield further insights.

We do agree that the results can, and should be visualized in a more interesting way. Figures 4-7 are now included as supplemental figures, as we find that it is still relevant to present the specific results for the interested reader. In the revised manuscript we have now included two figures presenting the phylum and genus taxonomical levels and combining the non-inflamed ileum, non-inflamed colon, inflamed colon and fecal samples of responders and non-responders at the different timepoints. In these figures the overall microbial abundances are also presented throughout the study to get more basic characteristics of the microbiota composition.

5. When writing results that have the reader compare across plots, it would be helpful to keep the same y-axis across the set of plots. One example is pane 3C, which would be easier to see the key results if all Y axes were $10^7 - 10^{12}$

This has now been corrected in the figure. However, as the copy numbers are differing between the mucosal and fecal samples, we decided to keep 10^8 in the mucosal samples and 10^{12} in the fecal samples. As we now have replaced figures 4-7 we hope this is easier to read when the difference is present only in one figure.

6. Y axis labels should be more descriptive, so it is easy to know what the sample is and what the units are (copy number / gram stool). Right now everything just says copy number, which is not very helpful.

Thank you for the comment, this has now been corrected in the revised manuscript.

Re: Spectrum01894-24R1 (**The total gut mucosal and fecal bacterial load increases in successful treatment of inflammatory bowel disease with infliximab**)

Dear Dr. Rebecka Ventin-Holmberg:

Your manuscript has been accepted, and I am forwarding it to the ASM production staff for publication. Your paper will first be checked to make sure all elements meet the technical requirements. ASM staff will contact you if anything needs to be revised before copyediting and production can begin. Otherwise, you will be notified when your proofs are ready to be viewed.

Sincerely,
Hao-Yu Liu
Editor
Microbiology Spectrum